# Role of Alpha-Fetoprotein (AFP) in Diagnosing Childhood Cancers and Genetic-Related Chronic Diseases

**DOI:** 10.3390/cancers15174302

**Published:** 2023-08-28

**Authors:** Joanna Głowska-Ciemny, Marcin Szymanski, Agata Kuszerska, Rafał Rzepka, Constantin S. von Kaisenberg, Rafał Kocyłowski

**Affiliations:** 1PreMediCare Prenatal Research Center, ul. Czarna Rola 21, 61-625 Poznań, Poland; md.marcinszymanski@gmail.com (M.S.); kuszer@o2.pl (A.K.); biuro@premedicare.pl (R.K.); 2New Med Medical Center, ul. Szamotulska 100, 60-566 Poznań, Poland; 3Department of Gynecology and Obstetrics, Institute of Medical Sciences, University of Zielona Gora, ul. Zyty 28, 65-046 Zielona Góra, Poland; r.rzepka@inm.uz.zgora.pl; 4Department of Obstetrics and Gynecology, Hannover Medical School, Carl-Neuberg-Str. 1, D-30625 Hannover, Germany; vonkaisenberg.constantin@mh-hannover.de

**Keywords:** AFP, genetic syndrome, neonatal tumor, pediatric oncology, diagnostic pitfalls

## Abstract

**Simple Summary:**

The article presents the role of alpha-fetoprotein in the diagnosis and monitoring of treatment for selected genetic diseases and early childhood cancers. The authors draw attention to diagnostic pitfalls related to physiological AFP production in the first year of life, inconsistencies in laboratory tests, and result interpretation.

**Abstract:**

Alpha-fetoprotein (AFP) is a protein commonly found during fetal development, but its role extends beyond birth. Throughout the first year of life, AFP levels can remain high, which can potentially mask various conditions from the neurological, metabolic, hematological, endocrine, and early childhood cancer groups. Although AFP reference values and clinical utility have been established in adults, evaluating AFP levels in children during the diagnostic process, treatment, and post-treatment surveillance is still associated with numerous diagnostic pitfalls. These challenges arise from the presence of physiologically elevated AFP levels, inconsistent data obtained from different laboratory tests, and the limited population of children with oncologic diseases that have been studied. To address these issues, it is essential to establish updated reference ranges for AFP in this specific age group. A population-based study involving a statistically representative group of patients could serve as a valuable solution for this purpose.

## 1. Introduction

The half-life of AFP in neonatal and infant populations has been determined to be 5.5 days at birth, 11 days between 14 and 30 days after birth, and 33 days up to 4 months of age. The rate at which AFP degrades depends on factors such as birth weight, feeding method, gestational age at birth, and the additional production of this fetal protein by the neonatal liver after delivery [1] (see Table 1 and Figure 1). Immediately after delivery, newborns typically exhibit alpha-fetoprotein (AFP) concentrations ranging from approximately 17,200 to 44,300 ng/mL [1,2]. However, prematurely born neonates tend to have an average concentration of 158,125 ng/mL [2]. The higher AFP level in premature neonates is attributed to their lower body weight and a less pronounced dilution effect. Over the course of the first 12 months after birth, the infant’s AFP concentration gradually decreases [1,2]. Pediatricians should bear in mind that the AFP levels can be high in first year of life and not implement hasty oncological diagnostics. Generally, the most significant decline occurs within the first 8 months postpartum, after which the AFP levels reach values typical for adults, with a maximum of 10–15 ng/mL in the serum [1,2].

## 2. AFP—Diagnostic Difficulties in Pediatrics

The precise definition of reference ranges and clinical utility of AFP in adults contrasts with the ambiguity surrounding its application in pediatric oncology. Studies conducted in pediatric oncology since the 1970s have employed varying reference standard ranges for AFP and lack stringent reference values for biological materials other than serum, such as cerebrospinal fluid (CSF) [3]. Several factors contribute to this situation, which are outlined below.

### 2.1. Postpartum AFP Concentrations

Differentiating pathological AFP concentrations in newborns poses significant challenges due to the ongoing high physiological production of AFP during the fetal period. The AFP level produced by a neoplasm may not exceed the physiological concentrations observed in infancy. Therefore, it becomes particularly challenging to exclude a neoplastic process, especially in children born prematurely and up to 4 months of age [3,4].

### 2.2. Methods of Determination and Establishment of AFP Reference Values

Current recommendations in pediatric oncology continue to rely on AFP reference ranges that are based on tests conducted over 40 years ago, utilizing RIA tests calibrated to internal standards. However, the diagnostic tests employed today yield results that are essentially incomparable to those obtained 20–40 years ago. Furthermore, due to the lack of studies employing modern tests on newborns, the veracity of the previously reported high physiological values for the neonatal period remains uncertain.

### 2.3. Comparing the Results of AFP Concentrations

Currently, the harmonization of diagnostic tests for AFP among laboratories continues to rely on the 1975 WHO international standard. The test results are reported in international units (IU). However, the prevailing practice is to convert these results into micrograms per liter or milliliter (µg/L or µg/mL). It is usually assumed that 1200 ng of AFP corresponds to 1000 IU [5]. However, the interpretation of this conversion can vary depending on the specific test, leading to significant variability in results. In fact, some laboratories even dispute the validity of this standard altogether [3].

### 2.4. Size of Study Groups

Setting reference standards for AFP in pediatric oncology is challenging due to the limited number of cases available for testing. The rarity of cancer in children makes it difficult to obtain data from a larger population, resulting in a reliance on case reports and small case series. As a consequence, the reliability of reference ranges for AFP in pediatric oncology is relatively low. It may be more practical to focus on evaluating the dynamics of changes in AFP concentrations, which tend to increase for primary lesions and gradually decrease for recurrent tumors. Additionally, assessing the half-life of AFP, which is prolonged in tumors, can provide valuable information [6]. In pediatric oncology, diagnostic imaging and histopathology should be prioritized as more reliable methods, while AFP evaluation serves a complementary role. Nonetheless, there are selected genetic syndromes and pediatric cancers in which AFP holds the potential for diagnostic, monitoring, and prognostic applications (Figure 2).

## 3. Liver

### 3.1. Ataxia Telangiectasia (AT)

The condition occurs with a frequency of 1:40,000 to 1:100,000 live births, making it the second most common autosomal recessive ataxia in children, following Friedrich ataxia. The underlying cause is a mutation in the ATM gene located on chromosome 11 at locus q22–23, which is responsible for DNA repair and regulation of the cell cycle by controlling the synthesis of the suppressor protein TP53. In 90% of patients, AFP levels are elevated, which distinguishes it from Friedrich ataxia [7,8]. This was demonstrated by Waldmann et al. in the 1970s, who conducted a study involving parents and siblings of children with AT [7,8]. AFP testing is not feasible in AT patients until they reach 2 years of age, primarily due to the high physiological levels of AFP in infants and the postnatal decline dynamics. Carrier individuals with the mutated gene typically have normal AFP levels, while AFP levels are high and tend to increase with age in AT patients [9]. There are several hypotheses regarding elevated AFP levels in certain conditions. The first hypothesis suggests that the increase in AFP is associated with progressive liver damage. The second hypothesis focuses on the role of the suppressor protein TP53, which plays a role in DNA damage repair and also acts as a repressor of the gene responsible for AFP synthesis during liver development and regeneration. When TP53 is deficient due to ATM mutations, AFP levels rise. A third hypothesis suggests that AFP synthesis increases in response to the damaged CNS’s need for building blocks for cell membranes during the process of myelination. AFP serves as a carrier protein for polyunsaturated fatty acids (PUFAs), which are essential for this purpose [9,10,11,12].

### 3.2. Primrose Syndrome

The condition occurs with a frequency of 1:1,000,000 births and follows an autosomal dominant inheritance pattern. It is associated with a de novo mutation in the ZBTB20 gene, which leads to a microdeletion at locus 3q13.31. The ZBTB20 gene acts as a key repressor of DNA transcription during birth and is responsible for various processes such as neurogenesis, fetal liver development, cell growth, detoxification, and glucose metabolism. The clinical presentation of the condition includes intellectual disability, macrocephaly, high postnatal growth, cataracts, deafness, auricular calcifications, and myopathy. The high levels of AFP observed in this condition are a consequence of the mutation in the ZBTB20 gene, which leads to the unblocking of AFP synthesis in the liver (normally, the gene acts as a repressor) [13].

### 3.3. Type I Tyrosinemia

The condition is found in approximately 1:100,000 births and follows an autosomal recessive inheritance pattern due to a mutation in the FAH gene located on chromosome 15q23–q25. This mutation results in a deficiency of fumarylacetoacetase hydrolase, leading to the accumulation of toxic tyrosine metabolites, namely fumarylacetoacetate and maleylacetoacetate, in the liver and kidneys. These metabolites have mutagenic properties and inhibit porphobilinogen synthesis, leading to porphyria-like seizures [14].

In the early form of the disease (<2 months of age), there is acute liver failure, which carries a high mortality rate. High levels of AFP, a marker of early liver regeneration that begins in fetal life, are particularly observed in this form. In the late form (occurring after 6 months of age), cirrhosis, hypophosphatemic rickets, and liver failure are typical. With time, the risk of liver cancer increases; it occurs in 37% of patients over 2 years of age, typically between 4 and 5 years of age. The most common form is hepatocellular carcinoma (HCC), but hepatoblastoma (HB), as well as mixed types, can also occur. AFP serves as an early marker of hepatic remodeling, neoplastic transformation, and porphyria seizures in this condition [15,16,17]. According to Koelink et al., not only an increase in AFP levels but also a sustained steady level with a weak downward trend can predict the onset of HCC [18]. Long-term administration of nitisinone (2-(2-nitro-4-3 trifluoro-methylbenzoyl)-1,3-cyclohexanedione—NTBC), a well-established therapy for patients with type I tyrosinemia, is recommended. NTBC reduces the risk of HCC and porphyria attacks and leads to a decrease in AFP levels [19,20]. However, Bhushan et al. observed in their patient cases that even with long-term NTBC therapy and normalization of AFP, the risk of HCC is not completely eliminated [16,21].

### 3.4. Progressive Familial Intrahepatic Cholestasis—PFIC2

The condition occurs with a frequency of approximately 1:50,000–1:100,000 births and follows an autosomal recessive inheritance pattern. The underlying mutation affects the ABCB11 gene, located on chromosome 2 (2q24), which is responsible for encoding the BSEP protein. BSEP is a membrane transport protein found on the surface of hepatocytes. The mutation disrupts the normal transport of bile from hepatocytes to the bile ducts, resulting in its accumulation within hepatocytes. This leads to chronic inflammation and carcinogenesis. Patients with this condition typically present with jaundice and increased pruritus either in the newborn period or early childhood. It accounts for approximately 10–15% of cases of neonatal cholestasis. The disease progresses rapidly, resulting in cirrhosis, HCC (5–15% of children with PFIC-2), biliary tract cancer, and liver transplantation. AFP is a marker of increased disease progression to neoplasia [17,22,23].

### 3.5. Neonatal Intrahepatic Cholestasis Caused by Citrin Deficiency—NICCD

NICCD is an autosomal recessive disorder resulting from a mutation in the SLC25A13 gene, which is responsible for encoding the citrin protein. Citrin is a mitochondrial transport protein involved in the urea cycle. A deficiency of citrin leads to serum hyperammonemia as well as abnormalities in glycolysis and beta-oxidation of fatty acids. As a result, hepatocytes are unable to utilize glucose and fatty acids as an energy source, leading to hyperlipidemia and hepatic steatosis. The condition is most commonly reported in East Asian countries such as China, Japan, and Korea, but cases have been documented worldwide. Neonatal cholestasis, accompanied by low birth weight and elevated AFP levels, is a characteristic feature. Despite these symptoms, NICCD is generally considered a benign disease, as it tends to spontaneously resolve within the first year of life with the introduction of lactose-free and/or medium-chain fatty acid (LF/MCT) nutrition. However, if left untreated, it can progress to cirrhosis, necessitating liver transplantation [24,25,26].

### 3.6. Transaldolase Deficiency—TALDO

The deficiency of transaldolase, caused by a mutation in the transaldolase gene located at 11p15.5–p15, is an autosomal recessive disorder that occurs with a frequency of 1:1,000,000 births. Transaldolase is an enzyme produced in the liver that plays a role in the pentose phosphate pathway. Deficiency of this enzyme results in a defect in the pentose phosphate pathway and the accumulation of polyols in the blood, urine, and CSF. Elevated AFP levels serve as a marker of liver regeneration and tumorigenesis [27]. In a study by Rodan et al., the administration of N-acetylcysteine, a precursor of glutathione, resulted in the normalization of AFP levels and a reduced risk of HCC in later life. N-acetylcysteine was shown to improve beta-catenin phosphorylation, which blocks carcinogenesis [28]. On the other hand, Lipinski et al. observed a spontaneous decrease in AFP levels with increasing age in TALDO patients without the introduction of any specific treatment [17,28,29,30].

### 3.7. Hepatitis B (HBV) in Children

The infection is primarily transmitted vertically. Approximately 90% of newborns from HbsAg and HbeAg-positive mothers will develop chronic infections if they do not receive postnatal immunoprophylaxis. In contrast to adults, where only 5–10% develop chronic hepatitis when infected, 25–90% of infected newborns experience chronic hepatitis. To predict the occurrence of HCC, determining the levels of AFP is useful. Elevated AFP levels typically coincide with increased aminotransferases and the presence of fibrosis, indicating necro-inflammatory changes. Elevated AFP levels can also be observed in asymptomatic chronic carriers with normal aminotransferase values. It is important to consider the possibility of HBV infection in patients with other tumors that secrete AFP to avoid unnecessary surgical interventions and chemotherapy [31]. A small percentage (0.01–0.03%) of children who are chronic HBV carriers will develop HCC before reaching adulthood. Kim et al. demonstrated that individuals with high AFP levels (>100 ng/mL) who are not diagnosed with HCC often have either HBV or HCV infections. Furthermore, persistently high AFP levels for over a year despite antiviral treatment significantly contribute to the development of HCC [32]. HBV-related HCC in children predominantly affects males, occurs later in life, and tends to be more aggressive compared to HCC caused by other factors. In countries where neonatal HBV vaccination is implemented, the incidence of hepatitis and liver cancer related to HBV has been dramatically reduced [17,33,34].

## 4. Hematopoietic System

### Fanconi Anemia

The condition occurs with a frequency of 1–5/1,000,000 births and is inherited in an autosomal recessive manner. The genetic basis of the condition involves numerous mutations, approximately 19 in total, that affect genes responsible for DNA repair. A diagnostic feature is the instability of chromosome structure following exposure to alkylating drugs. Aslan et al., based on serial measurements of serum AFP levels in pregnant carriers, demonstrated that AFP cannot serve as a marker for amniotic fluid in prenatal diagnosis, as even pregnant women carrying affected fetuses exhibit typical serum AFP levels [35]. In this disease, AFP levels are elevated from birth, remain constant, and are independent of concurrent liver conditions and androgen treatment (originally, AFP measurement was used to detect liver adenomas resulting from androgen treatment). Studies have shown that other bone marrow disorders with a genetic basis, such as Blackfan–Diamond syndrome, Shwachman–Diamond anemia, or congenital dyskeratosis congenital, exhibit normal AFP levels [36]. Until recently, AFP was used as a simple diagnostic tool for Fanconi anemia (FA), with studies indicating varied sensitivity: 93% sensitivity and 100% specificity (Cassinat et al., 2000) [37]; 46% sensitivity (Aslan et al., 2002) [19]; and 71% sensitivity (Salem et al., 2019) [17]. However, a recent study by Alter et al. suggests a sensitivity of approximately 25%, although their study included a group of FA patients with a higher median age [36]. The primary treatment for FA is bone marrow transplantation, which does not completely normalize AFP levels, but may slightly reduce them [29,30,38]. Regarding the source of elevated AFP in patients, there is uncertainty. Studies by Salem et al. and Blanche et al. demonstrated significantly higher AFP levels in patients with FANCD1/BRCA2 mutations compared to other types of mutations [36,38]. Aslan et al. propose that impaired postnatal suppression of the AFP gene and/or a shift in production from AFP to albumin may be contributing factors [35]. It is also believed that multipotent progenitor cells in the bone marrow play a role. Two subtypes are recognized: fetal hepatic stem/progenitor cells (FHSC) and intrinsic hematopoietic stem/progenitor cells (HSPC), as well as bone marrow mesenchymal cells that can differentiate into hepatic stem cells and migrate to the liver when it is damaged. HSPCs can also migrate to the liver and serve as precursors to oval liver stem cells [37,39].

## 5. Endocrine System

### Hypothyroidism

The level of AFP during fetal life is influenced by the levels of thyroid hormones. This is because thyroid-stimulating hormone (TSH) binds with AFP in the fetal blood plasma. It is believed that triiodothyronine (T3) plays a role in the transcriptional switch from AFP to albumin early in life. In the absence of T3 (hypothyroidism), this physiological process is delayed [40]. T3 has been shown to induce the differentiation of hepatic oval cells (HOC) into hepatocytes in the rat liver. It is worth noting that only HOC in the liver produces AFP during infancy [7,41]. A laboratory study demonstrated that the level of AFP decreased in mice treated with thyroxine (T4). Conversely, in cases of congenital hypothyroidism, AFP levels increase alongside elevated TSH levels and low T4 levels. These elevated AFP levels persist after birth, whereas in healthy children, AFP levels decrease rapidly. This is attributed to the prolonged half-life of AFP, which extends to 12 days (typically 5–6 days), as a result of its impaired breakdown rate in the liver due to low T4 levels [1]. AFP is also used as a diagnostic marker for ovarian tumors in van Wyk–Grumbach syndrome, when long-term untreated hypothyroidism in children leads to precocious puberty [42,43,44].

## 6. Cancers

In the field of pediatric oncology, AFP is utilized as a diagnostic tool and for monitoring the effectiveness of surgical treatment and chemotherapy in cases of embryonal HB, HCC, and germ cell tumors (GCTs) [45].

### 6.1. Hepatoblastoma

HB is the most common malignant liver tumor in children, accounting for 67–80% of cases and occurring at a rate of 1–10/1,000,000 births. It represents approximately 1–2% of pediatric cancers [46]. The tumor is predominantly localized in the right lobe of the liver, as observed in 55–60% of cases [4]. Less than 10% of HB cases develop prenatally, and the average age of diagnosis is 18 months. HB is more frequently diagnosed in premature infants, particularly those with birth weights below 1500 g [47]. Prematurity poses challenges for diagnosis due to the typically high levels of AFP in this group of newborns compared to full-term babies. Histopathologically, HB is classified into epithelial types (including fetal, embryonal/fetal, macrotrabecular, and small cell undifferentiated) and epithelial-mesenchymal types (two subtypes). Some HB subtypes, such as fetal and undifferentiated small cells, exhibit normal AFP levels [45,48]. Only around half of HB cases show elevated AFP levels above the upper reference values, as tumor-derived AFP levels are often not significantly higher than the physiologically high levels observed during the first months of life [4]. In the past, AFP was considered to have prognostic value, with very high (>1,000,000 ng/mL) or very low (<100 ng/mL) levels indicating poor prognosis in HB [49]. However, the Children’s Hepatic Tumors International Collaboration (CHIC), after analyzing its databases and identifying SMARCB1 mutations in a group of HBs with the small cell undifferentiated subtype or HBs with low AFP levels (with survival rates of 24–37.5%), concluded that some of these HBs were rhabdoid tumors with an extremely unfavorable prognosis (3-year OS—0%). After excluding rhabdoid tumor cases from the analyzed group, it was found that the presence of the small cell undifferentiated subtype or low AFP levels no longer had poor prognostic significance [49]. After surgery, AFP levels are expected to decrease below the upper reference range, and failure to do so indicates unsuccessful tumor resection or early recurrence. Similarly, during chemotherapy, a slow decline in AFP levels indicates an unfavorable prognosis. Following the completion of treatment, an increase in AFP levels above age-specific reference values, even in the absence of clinical and imaging evidence of the tumor, suggests recurrence. Currently, there are insufficient data to determine whether there is a correlation between AFP levels at diagnosis and relapse after complete remission (CR). A study by Li et al. showed that AFP levels >1000 ng/mL at diagnosis are not an independent prognostic factor for relapse after CR, as there was no statistically significant difference in relapse-free survival (RFS) (data were inconclusive for cases <100 vs. >100) [45,48,50].

### 6.2. Hepatocellular Carcinoma

HCC is the second most common malignant liver tumor in children, accounting for 2–33% of cases. It occurs at a rate of 0.41/1,000,000 births. Several predisposing factors contribute to its development, including the vertical transmission of HBV, tyrosinemia, progressive familial intrahepatic cholestasis, glycogen storage diseases, Alagille syndrome, congenital portal-systemic shunts, Wilson disease, alpha-1-antitrypsin deficiency, transaldolase deficiency, Gardner’s syndrome, FA, ataxia-telangiectasia, familial adenomatous polyposis, and primary sclerosing cholangitis [51] (see Table 2). HCC primarily affects children over 5 years of age and can arise in the presence or absence of de novo cirrhosis, sometimes associated with underlying liver diseases. Histopathologically, HCC can be classified as conventional HCC, fibrolamellar HCC, or HCC with HB elements [52]. On average, approximately 50% of HCC cases exhibit elevated AFP levels [51]. In the fibrolamellar form, only 10% of cases have elevated AFP levels. High AFP levels are associated with higher mortality rates [52]. Surgical treatment, including complete tumor resection (possible in only 30% of diagnosed cases) along with additional chemotherapy or liver transplantation, is the standard approach for managing HCC [17,39,52,53].

### 6.3. Germ Cell Tumors

GCTs account for 3.5% of cancers in children up to the age of 15 and 13.9% in the 15–19 age group [54]. They are characterized by male dominance, with the exception of SCT. GCTs encompass a group of neoplasms derived from pluripotent germ cells, including both benign and malignant tumors. They can occur within the gonads (50% of cases up to age 4) as well as outside the gonads (50% of cases up to age 4, and 10–15% after puberty) [55]. These tumors are commonly found in midline locations of the body, such as the sacrococcygeal region, mediastinum, skull (pineal region), retroperitoneal space, nasopharynx, orbit, neck, uterus, and vagina [56]. The prevailing hypothesis is that their presence in these locations results from the misplacement of primordial germ cells (PGCs) during their migration from the yolk sac to the genital ridges, from which the definitive gonads develop. Aberrant migration leads to the ectopic localization of germ cells along the midline of the body. Malignant transformation of these cells in extragonadal sites gives rise to GCTs [56]. Elevated levels of AFP in these tumors are attributed to the presence of immature or malignant tissue elements derived from the yolk follicle [55,56]. The combination of AFP and beta-HCG can detect 5–60% of GCTs, depending on the histopathological subtype, with detection rates reaching up to 85% in extracranial localizations. Only 20% of early-stage GCTs exhibit elevated AFP levels [57,58]. According to Erlich et al., AFP alone detects 10–60% of nonsquamous GCTs [59]. The prognostic value of AFP at the time of GCT diagnosis remains controversial. Frazier et al., in a summary of seven trials conducted by the Children’s Oncology Group and the Children’s Cancer and Leukemia Group, suggested that AFP levels above 10,000 ng/mL at diagnosis were associated with a worse prognosis, indicated by lower event-free survival (EFS) and overall survival (OS), although statistical significance was not achieved (*p* = 0.45) [5,60]. In contrast, Freseneau et al., in the TGM 95 study, reported that baseline AFP levels did not affect 5-year recurrence-free survival (5y-RFS) [61]. In adults with malignant GCTs, the decrease in AFP during treatment is an important prognostic factor [61]. However, in children, there is no consensus in the literature regarding the prognostic significance of AFP normalization. According to the French TGM 95 study conducted by Freseneau et al., the predicted time to normalization of AFP did not have significant prognostic value [61]. On the other hand, a study by Faure-Conter et al. on the TGM13-NS protocol, which aimed to achieve high cure rates with minimized chemotherapy doses in children with GCT, demonstrated that AFP normalization had a prognostic impact on EFS (HR = 1.003 [1000–1007]) [62]. O’Neill et al., analyzing data from the Children’s Oncology Group (COG) AGCT0132 protocol, showed that children who had a satisfactory decrease in AFP (with normalization of any of the first two measurements more than 7 days after starting chemotherapy or an AFP half-life of ≤7 days) had a lower cumulative 3-year recurrence rate compared to those with an unsatisfactory decrease (11 vs. 38%) [63]. AFP can serve as a useful tool for diagnosing GCT recurrences; however, it cannot be solely relied upon as a diagnostic measure. In a study conducted by Trigo et al., it was found that 68% of patients with recurrence had elevated levels of the marker (AFP or beta-HCG) at both initial diagnosis and recurrence [64]. Conversely, Keskin et al. demonstrated no significant disparity in AFP levels between patients with and without GCT recurrence, despite receiving the same treatment regimen. The only distinction observed was in the AFP half-life [65]. Nevertheless, it is important to note that not all GCTs secrete AFP, and some present challenges in terms of localization through biopsy. Therefore, a more precise alternative appears to be emerging in the form of microRNAs (miR-371a-3p/-5p—373 and miR-302/367), which offer simplicity and ease of use for diagnostic purposes. These microRNAs are short single-stranded fragments of noncoding RNA responsible for regulating gene expression by influencing translation blocking or mRNA degradation. Alterations in microRNA expression contribute to the initiation of carcinogenesis [66]. In terms of diagnosis and recurrence monitoring, miRNA demonstrates higher specificity (93.4%) and sensitivity (88.7%) compared to AFP, regardless of the histopathological type, patient age, or anatomical location of the tumor [57,67,68].

### 6.4. Intracranial Germ Cell Tumors (IC-GCTs)

IC-GCTs account for 0.3–3.4% of childhood CNS tumors in North America and Europe, but the incidence rises to 15% in East Asia [69]. IC-GCTs can be categorized into two main types: germinomas (GER) and nongerminoma GCTs (NG-GCT), which include YST, EC, choriocarcinomas, teratomas, and mixed GCTs [70]. Intracranial teratomas are the most common, followed by immature teratomas. They can present in various forms, ranging from large tumors causing mass effects and spreading into the nasopharynx and orbit to smaller lesions causing hydrocephalus. The most common sites of origin are around the pineal gland, the Turkish saddle, and the third ventricle. In approximately one-third of cases, the starting point cannot be determined due to the mass effect. To avoid the need for a biopsy, determination of AFP and b-HCG levels in CSF and serum can be performed. If AFP is confirmed to be >25 ng/mL and b-HCG is confirmed to be >50 IU/L in at least one sample of serum or fluid, a biopsy can be avoided. According to the International Society of Pediatric Oncology (SIOP) study, serum and/or CSF AFP levels ≥25 ng/mL and/or b-HCG levels greater than or equal to 50 IU/L indicate a diagnosis of NG-GCT. The Children’s Oncology Group (COG) suggests cutoff points of 10 ng/mL for AFP and 100 IU/L for b-HCG [71,72]. In a study by Sathisamitphong et al. involving 63 IC-GCT cases, there was an 84.3% concordance between serum AFP and CSF levels [73]. Frappaz et al. reported that AFP levels are higher in serum compared to CSF, whereas b-HCG levels are comparable [71]. Legault et al. demonstrated that AFP levels are highest in serum, followed by CSF obtained through a lumbar puncture, and finally, CSF obtained through a ventricular puncture [74]. Takami et al. showed that the sensitivity of individual markers in detecting NG-GCT CNS is as follows: b-HCG (>100 IU/L) at 61.5%, AFP (>10 ng/mL) at 83.3%, and both markers together at 94.7% [75]. If these two markers are negative, a biopsy is required to differentiate between teratoma and germ cell carcinoma, as the treatment regimens differ [70]. In cases where there is no admixture of YST within the tumor mass, high levels of AFP are attributed to the presence of immature elements from glandular epithelium characteristic of the gastrointestinal tract and/or ependyma (a type of glial tissue) lining the ventricular system, and occasionally structures resembling liver tissue. Hong et al. demonstrated that only the determination of AFP (≥10 ng/mL) along with beta-HCG (≥50 IU/mL) has prognostic value for NG-GCT or malignant GCT, indicating worse EFS and OS compared to the determination of a single marker alone [76]. In the SIOP-CNS-GCT-96 trial, patients with serum AFP >1000 ng/mL and/or CSF post-treatment with the established residual disease had worse progression-free survival [72]. Among patients with recurrence, those who had AFP in serum or CSF ≤25 ng/mL had a better prognosis [56,77,78].

### 6.5. Malignant Saccrococygeal Germ Cell Tumor

The occurrence of SCT has a frequency of 1 in every 35,000–40,000 births. The most common type is mature teratoma; however, 11–35% of cases involve a mixture of a malignant component, most commonly YST or EC [60]. Unlike other GCTs, this condition is more prevalent among girls. It presents as the most common tumor in newborns, typically as an outward-growing mass. It can also manifest after infancy, typically before the age of 3, with an inward-growing growth pattern characterized by buttock asymmetry and gastrointestinal and/or urinary tract issues, as well as lower limb dysfunction. The risk of malignant transformation increases with the child’s age, from 11 to 35% at birth to over 70% in cases diagnosed after >1 year of age. The primary treatment is surgical intervention, with the addition of chemotherapy for malignant cases. The use of AFP for detecting prenatal lesions is not reliable, as it does not consistently elevate in maternal serum in most cases, regardless of whether the tumor is mature or immature or whether it is covered by skin [56]. However, AFP is valuable as an early marker for assessing the completeness of tumor resection as well as for monitoring malignant recurrence after initial surgery (75% of recurrences show elevated AFP) and chemotherapy (monitoring for up to 3–5 years post-treatment) [79]. This is because teratomas prone to malignancy often contain a mixture of YST (present in 22–56% of recurrences). It has been demonstrated that AFP has a prolonged half-life in SCT tumors with a tendency for recurrence (with YST admixture) and in immature teratomas (without YST admixture). AFP levels have prognostic significance [56,80,81,82].

### 6.6. Special Histopathological Cases of GCT Connected with High AFP Levels

#### 6.6.1. Yolk Sac Tumor (YST = Endodermal Sinus Tumor)

YST is the most common malignant germ cell tumor in children and is histologically composed of yolk sac mesenchymal cells. Approximately 70–90% of YST cases secrete AFP [83,84], but in some instances, metastatic lesions or treatment remnants may lose the ability to produce AFP [85]. It primarily occurs in the male and female gonads [55,56]. In boys, it is the leading cause of testicular cancer, while in girls, it is a rare tumor of the ovary [55]. YST can occur independently in about 60% of cases, mainly in the pediatric population, or as a component of other GCTs, most commonly teratomas or dysgerminomas, accounting for 40% of cases in the postpubertal age group [56,86]. Approximately 15% of YST lesions can occur extragonadal in midline organs of the body, such as the CNS, paranasal sinuses, bladder, vagina, prostate, and retroperitoneal space [83] (known as extragonadal germ cell tumors—EGGCT). It can also manifest in the liver, where it must be differentiated from HB, as both tumors can exhibit high levels of AFP. Another extramedian site of occurrence is the kidneys, which can sometimes be mistaken for a Wilms tumor [87]. Except for AFP, no other feature of the preoperative clinical examination differentiates these two tumors [88]. However, AFP is not produced in the kidneys during prenatal or childhood stages, and thus AFP has never been a marker for Wilms tumor. There are rare situations in which Wilms tumors consist of tissue resembling nephroblastoma as well as tissue morphologically corresponding to a teratoma, leading to AFP production [89,90,91]. In giant mixed GCTs, small foci of YST, the source of AFP, may be accidentally overlooked during the examination of tumor samples [56,92]. AFP is used for monitoring and evaluating the effectiveness of treatment but is not helpful for prognosis [54]. A meta-analysis by Guo et al. on the prognostic value of AFP in ovarian YST demonstrated that only postoperative AFP values are useful. High postoperative AFP levels were associated with worse OS (OR = 0.16, 95% CI: 0.05–0.48) and RFS (OR = 0.18, 95% CI: 0.08–0.43) compared to low postoperative AFP levels in ovarian yolk sac tumor (OYST) patients [93]. De la Motte Rouge et al. showed that an early decline in AFP levels during chemotherapy for OYST predicts better OS (100%) compared to an unfavorable decline (OS 49%, 95% CI: 26–72%) [94]. Currently, a more sensitive marker, ZBTB16 (Zinc finger and BTB ((Broad/complex/Tramtrack/Bric a Brac) domain-containing 16)), is used, which can also detect extragonadal and metastatic YST lesions with a sensitivity of 91.6% [85].

#### 6.6.2. Embryonal Carcinoma (EC)

Histopathologically, germinoma is the most primitive form of germ cell tumor, capable of differentiating into YST or immature teratoma. It consists of cells that produce AFP (from YST) and b-HCG (from choriocarcinoma) [56]. Germinoma is a common component of mixed germ cell tumors (MGCT). According to a study by Ataikiru et al. involving Romanian children with MGCT, 87.5% of cases before puberty and 64.7% after puberty had an EC component [57]. It primarily affects males and is extremely rare in females [95]. Germinoma can occur in both the gonads and the CNS. In the CNS, it typically arises in the region of the third ventricle and pineal gland. The diagnosis can be improved by measuring B-HCG and AFP levels in both serum and CSF, as mentioned earlier [96].

## 7. Ovarian Sertoli-Leydig Cell Tumor (SLCT)

A rare, unilateral mixed-sex cord-stromal tumor. It is mainly found in young women (75% of cases), but the youngest known case was 9 months old [97]. Approximately 40–50% produce androgens (virilization symptoms), less often estrogens, and least often both types of sex hormones simultaneously [56,97]. The tumor is composed of Sertoli cells, Leydig cells, fibroblasts, and stromal cells in varying proportions, but may also have a component of glandular intestinal cells producing mature or immature hepatocytes, which are responsible for AFP production [98]. Another hypothesis is that Leydig cells and hepatocytes are morphologically similar. Still, other authors suggest that the source of AFP is the presence of cells similar to Leydig cells, but without the presence of typical crystals on histopathological examination [99]. Another AFP-producing component may be poorly differentiated tissue fragments of the endodermal sinus difficult to identify on histopathological examination (YST-like) [99,100,101].

## 8. Conclusions

Careful consideration should be given to using AFP levels as a basis for clinical decisions in neonatology since they are physiologically elevated from birth until the first year of life.

In young children, elevated AFP levels can mask the presence of certain genetic diseases, liver regeneration in chronic diseases, and tumorigenesis processes.

In pediatric cases, AFP remains a valuable marker for liver tumors and GCTs that involve tissue elements derived from the yolk follicle. Monitoring AFP levels in children with various chronic liver diseases can help predict the early onset of HCC.

It is necessary to develop new pediatric reference ranges for modern AFP diagnostic tests based on current laboratory techniques.

## Figures and Tables

**Figure 1 cancers-15-04302-f001:**
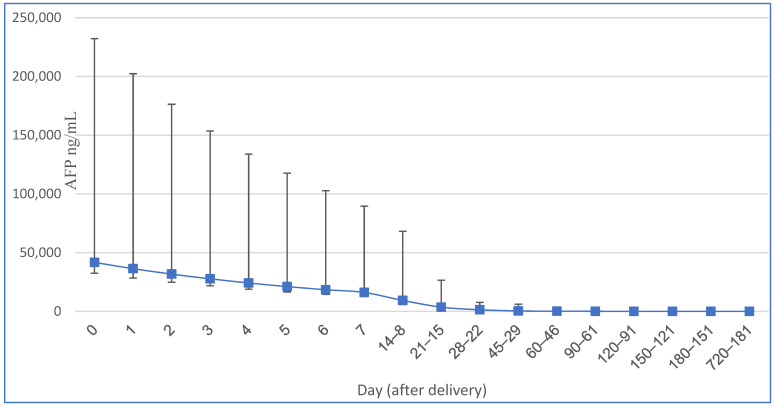
Serum AFP changes (and 95.5% interval) in term neonates modified from [2].

**Figure 2 cancers-15-04302-f002:**
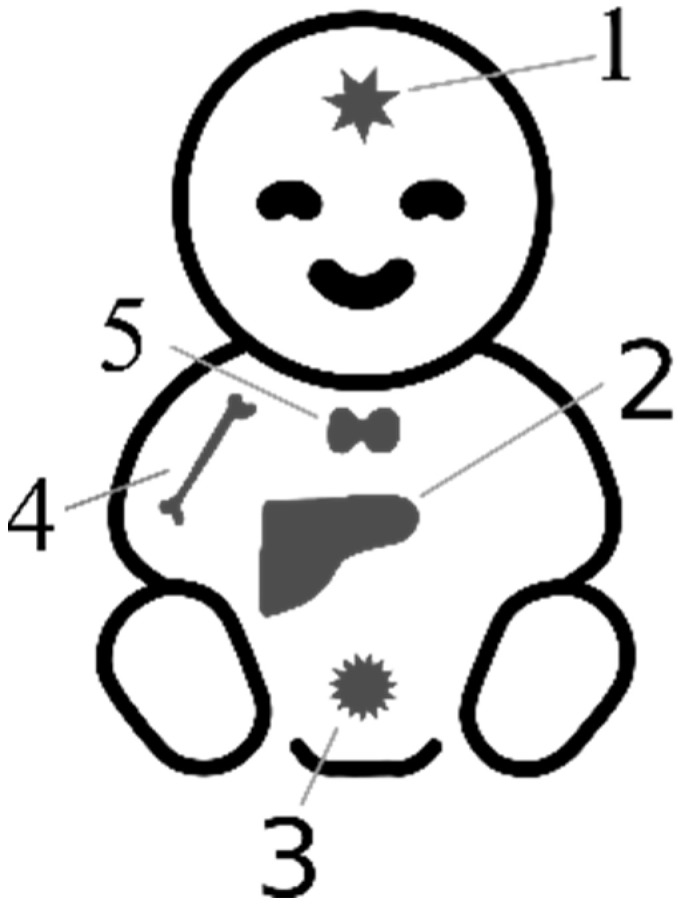
Sources of elevated AFP in infancy and childhood. 1—Intracranial germ cell tumors (IC-GCTs); 2—hepatoblastoma (HB), hepatocellular carcinoma (HCC), ataxia teleangiectasia (AT), primrose syndrome, tyrosynemia type I, neonatal intrahepatic cholestasis caused by citrin deficiency (NICCD), progressive familiar intrahepatic cholestasis (PFIC2), transaldolase deficiency (TALDO), hepatitis B (HBV); 3—malignant saccrococygeal GCT; 4—Fanconi anemia; 5—congenital hypothyroidism, van Wyk–Grumbach syndrome.

**Table 1 cancers-15-04302-t001:** Serum alpha-fetoprotein (AFP) levels in term neonates [2].

Neonatal Age (Days)	AFP Mean (ng/mL)	AFP 95.5% Interval (ng/mL)	Half-Life (Days)
0	41,687	9120–190,546	
1	36,391	7943–165,959	
2	31,769	6950–144,544	
3	27,733	6026–125,893	
4	24,210	5297–109,648	
5	21,135	4624–96,605	5.1
6	18,450	4037–84,334	5.1
7	16,107	3524–73,621	5.1
8–14	9333	1480–58,887	5.1
15–21	3631	575–22,910	5.1
22–28	1396	316–6310	5.1
29–45	417	30–5754	14
46–60	178	16–1995	14
61–90	80	6–1045	28
91–120	36	3–417	28
121–150	20	2–216	42
151–180	13	1.25–129	42
181–720	8	0.8–87	no correlation

**Table 2 cancers-15-04302-t002:** Common risk factors for hepatocellular carcinoma (HCC) in infancy and childhood.

Risk Factor with Elevated AFP	Risk Factor with Decreased or Normal AFP
Hepatitis B	Alpha-1 antitrypsin deficiency
Tyrosinemia	Glycogenosis
Progressive familiar intrahepatic cholestasis type 2	Parto-systematic shunts
Transaldolase deficiency	Alagille syndrome
Ataxia teleangiectasia	Gardner syndrome
Fanconi anemia	Familial adenomatous polyposis
Biliary artesia	Budd–Chiari syndrome

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
