# Peer review of "Role of Alpha-Fetoprotein (AFP) in Diagnosing Childhood Cancers and Genetic-Related Chronic Diseases"

_cancers, 2023, doi:10.3390/cancers15174302_

Round 1

Reviewer 1 Report

Although authors have done a lot of research, there is no real new material or insight into the role of paediatric cancer. Not having any knowledge of the genetic conditions I will not judge on those topics. Some statements are wrongly quoted (eg.  Paradoxically, the majority of cancers in children are diagnosed during the first year of life). Describing the tumors and their behavior are to extensive and not solely focused on the AFP levels, as this is the subject of the article. In my opinion this confuses the reader to much. The headings of GCT separate from YST and EC is strange, being one histological subtype of the other.  

The subject of the determination, metabolism and clinical relevance of AFP probably is very useful but I would advise to strictly keep to the AFP without elaborating to much of all the characteristics of the disease itself and to make sure that all statements are scientifically sound and correct. 

Probably not completely reliable, not being english native speaker myself, but I get the impression that some of the confusing sentences have their origin in the faulty english phrasing such as:  Consequently, it is not advisable to conduct unnecessary oncological diagnoses during this period. 

Reviewer 2 Report

The paper seems a Review rather than a new way of checking or understanding AFP role. The role of this paper is mainly pedagogic. In attachment some notes and observations

Round 2

Reviewer 2 Report

no

Part of the response is not in english